# Mild Hypophosphatemia-Associated Conditions in Children: The Need for a Comprehensive Approach

**DOI:** 10.3390/ijms24010687

**Published:** 2022-12-30

**Authors:** Pablo Docio, Sandra Llorente-Pelayo, María Teresa García-Unzueta, Bernardo A. Lavin-Gómez, Nuria Puente, Fátima Mateos, Leyre Riancho-Zarrabeitia, Domingo Gonzalez-Lamuño, José A. Riancho

**Affiliations:** 1Servicio de Pediatría, Hospital U M Valdecilla, Instituto de Investigación Marqués de Valdecilla (IDIVAL), Universidad de Cantabria, 39011 Santander, Spain; 2Servicio de Análisis Clínicos, Hospital U M Valdecilla, Universidad de Cantabria, IDIVAL, 39011 Santander, Spain; 3Servicio de Medicina Interna, Hospital U M Valdecilla, Universidad de Cantabria, IDIVAL, 39011 Santander, Spain; 4Servicio de Análisis Clínicos, Hospital Sierrallana, 39300 Torrelavega, Spain; 5Servicio de Reumatología, Hospital Sierrallana, IDIVAL, 39300 Torrelavega, Spain

**Keywords:** hypophosphatemia, diabetes mellitus, thyroid disorders, short stature, GH deficiency

## Abstract

To better understand the causes of hypophosphatemia in children, we evaluated all serum phosphate tests performed in a tertiary hospital with unexpected but persistent temporary or isolated hypophosphatemia over an 18 year period. We collected 29,279 phosphate tests from 21,398 patients, of which 268 (1.2%) had at least one result showing hypophosphatemia. We found that endocrinopathies (n = 60), tumors (n = 10), and vitamin D deficiency (n = 3) were the medical conditions most commonly associated with mild hypophosphatemia, but in many patients the cause was unclear. Among patients with endocrinopathies, those with diabetes mellitus were found to have lower mean serum phosphate levels (mean 3.4 mg/dL) than those with short stature (3.7 mg/dL) or thyroid disorders (3.7 mg/dL). In addition, we found a correlation between glycemia and phosphatemia in patients with diabetes. However, despite the potential relevance of monitoring phosphate homeostasis and the underlying etiologic mechanisms, renal phosphate losses were estimated in less than 5% of patients with hypophosphatemia. In the pediatric age group, malignancies, hypovitaminosis D, and endocrine disorders, mostly diabetes, were the most common causes of hypophosphatemia. This real-world study also shows that hypophosphatemia is frequently neglected and inadequately evaluated by pediatricians, which emphasizes the need for more education and awareness about this condition to prevent its potentially deleterious consequences.

## 1. Introduction

Hypophosphatemia is a poorly recognized metabolic condition in clinical practice because serum phosphate is not part of routine biochemical screening in the primary care setting. In addition, phosphate results can be misinterpreted due to age-specific differences [1,2,3]. Moreover, the short-term effects of hypophosphatemia are usually mild and may go unrecognized, so mild hypophosphatemia is not considered relevant by many clinicians. Changes in serum phosphate levels may be caused by an abnormal glomerular filtration rate and reabsorption by the renal tubules, or by reduced intestinal absorption in relation to changes in intestinal transit time or inhibition of intestinal transport [2,4,5,6,7,8]. Transcellular phosphate shifts also impact serum phosphate levels. Thus, even in some conditions associated with depletion of the total amount of phosphate in the body, serum phosphate levels could be elevated due to the redistribution of phosphate into the intracellular space. 

Various endocrine disorders can be associated with persistent hypophosphatemia, which may have several consequences, particularly in the musculoskeletal system, including impaired bone mineralization, growth retardation, muscle weakness, and other manifestations of rickets [9]. That is the case in X-linked hypophosphatemic rickets and other less frequent disorders. If the manifestations of hypophosphatemia are not recognized and treated during childhood or adolescence, they may persist into adulthood, with a chronic and debilitating course that has a negative impact on patients’ quality of life [3].

The fact that mild hypophosphatemia frequently goes undetected, although it may have clinical consequences, motivated us to carry out a systematic search for cases of hypophosphatemia in children. 

## 2. Results

The retrospective and prospective arms included 29,279 phosphate results in 21,398 patients (Figure 1), of whom 268 (1.25%) had at least one result showing hypophosphatemia. Of them, thirty-six were not studied further, twenty-nine because of missing clinical and/or biochemical data, and seven because they had died by the time of the study.

Among the 232 patients with hypophosphatemia included in the study, 107 (46.1%) were males and 125 (53.9%) were females. The mean age of the first analysis showing hypophosphatemia was 5.8 years (6.0 years in the retrospective arm and 5.2 years in the prospective arm). The mean age at which each subject showed the lowest serum phosphorus was 6.8 years, with a median of 8. 

Overall, we identified 777 lab results showing hypophosphatemia, with a mean of 3.3 analyses per patient and a median of 2 (interquartile range, IQR, 1–4). In 77 patients (33.2%), serum phosphorus was determined only once, 61 of them in the retrospective analysis and 16 in the prospective arm. In 65.1% of patients with some phosphorus results below the reference range, hypophosphatemia was present in their first phosphate analysis available. Moreover, among patients with phosphorus analyzed twice or more, 44.5% presented hypophosphatemia in at least half of their samples. When assessing the lowest serum phosphate, the mean equivalent in the cohort was 3.6 mg/dL, with a median of 3.8 mg/dL (IQR 3.5–3.9) and a minimum value of 1.4 mg/dL. Other biochemical tests commonly obtained are shown in Table 1. 

When we analyzed all serum phosphate levels of each patient as described in the Methods Section, hypophosphatemia was classified as intermittent in one-hundred and twenty children (51.7%), isolated in one-hundred and eight (46.5%), and persistent in four (1.7%). 

After reviewing the clinical records, we found that 18 children (7.8%) had an underlying disorder that could be associated with hypophosphatemia (“expected hypophosphatemia”). Two other subjects were considered to present Metabolic Bone Disease of the Prematurity (MBDP). Finally, 212 patients were considered to present unexpected hypophosphatemia. The distribution across the subgroups with different time courses is shown in Figure 2.

The most frequent underlying conditions in patients with “expected” hypophosphatemia were malignant disease (eight patients, 3.4%), vitamin D deficiency (three patients, 1.3%), drug therapy (anticonvulsants, two patients, 0.9%; bisphosphonates, two patients, 0.9%; corticosteroids, two patients, 0.9%), and refeeding syndrome (one patient, 0.4%). 

A significant proportion of patients (25.4%) presented with endocrinopathies. The most common disorders were diabetes mellitus (21 patients, 9.1%), short stature (14 patients, 6%), thyroid disorders (13 patients, 5.6%), and others (precocious puberty: 2.6%; adrenal hyperplasia: 1.7%; and insulin resistance: 0.4%). Although some of them are known to be associated with hypophosphatemia in some circumstances, their role as a major factor causing hypophosphatemia in each patient was unclear. Therefore, we decided to analyze them separately, describing serum phosphate results in the most frequent groups, irrespective of whether they were considered “expected” or “unexpected”. The mean phosphate levels in children with hypophosphatemia associated with thyroid disorders were 3.7 mg/dL, 3.7 mg/dL in those with short stature, and 3.4 mg/dL in those with diabetes (Figure 3). An ANOVA test showed significant differences across the three groups (*p* = 0.024). The pair-wise comparisons showed statistically significant lower phosphate values in patients with diabetes, with differences that were close to the significance limits in the unadjusted test (*p*-values 0.049 and 0.044 for the comparisons of diabetes-short stature and diabetes-thyroid disorders, respectively), but nonsignificant after adjusting for multiple comparisons (*p*-values 0.15 and 0.13, respectively).

In the diabetes mellitus cluster, we identified 21 patients with 112 phosphorus determinations, of which 29 (26%) were below the reference range. The mean phosphorus value was lower in patients with glycemia over 200 mg/dL than in those with glycemia < 200 (3.8 vs. 4.3 mg/dL, respectively, *p* = 0.005), and there was a slight negative correlation between the serum levels of phosphorus and glucose (Pearson correlation coefficient: r = −0.26, *p* = 0.007; Figure 4a). Likewise, there was a negative relationship between phosphate and glycosylated hemoglobin (HbA1c; Pearson correlation coefficient: r = −0.62, *p* = 0.003) (Figure 4b). When patients were categorized according to their HbA1c levels, mean phosphorus was 4.4 mg/dL in subjects with HbA1c < 9 and 3.0 mg/dL in samples with HbA1c > 9 (*p* = 0.0004).

The group of thyroid disorders included one patient with congenital primary hypothyroidism, five with thyroiditis, four with subclinical hypothyroidism, and three with tertiary hypothyroidism. Among them, four patients had other associated endocrinopathies (diabetes in three; short stature in one). Overall, in the group of 13 patients with thyroid disorders and hypophosphatemia, concomitant serum phosphate and thyroid hormones were measured 76 times; in 15 samples (19.7%), serum phosphorus was below the reference range. We did not find a statistically significant correlation between T4 or TSH and phosphorus levels (*p* = 0.1 and *p* = 0.579, respectively).

The cluster of patients with short stature included one subject with GH deficiency secondary to craniopharyngioma and another patient with eosinophilic granulomatosis involving the CNS. Among them, seven out of fourteen (50%) presented with GH deficiency, and six of them received GH therapy. A total of 100 phosphorus determinations were obtained in these 14 patients, and 29 of them (or 29%) were below the reference range. There was no significant correlation between IGF-1 or IGFBP3 and serum phosphate (*p* = 0.20 and 0.85, respectively). However, it is important to note that in only 61% of the tests, phosphate and IGF-one measured simultaneously, and in only 27% of the cases, phosphate and IGFBP3 were assessed at the same time. When evaluating the group of subjects who were treated with GH, the mean phosphate levels were higher in the post-treatment samples than in the pre-treatment samples (4.7 ± 0.2 vs. 4.1 ± 0.2, *p* = 0.033).

Finally, we selected 10 patients with moderately to severe hypophosphatemia (<3.2 mg/dL) of unknown cause who were invited to attend the clinic for a more comprehensive clinical and biochemical study. After further studies, we found that serum phosphorus had normalized in seven cases; in two children, we found significant hypovitaminosis D; and in one case, hypophosphatemia persisted due to an unknown cause (a DNA sequence is pending to exclude a genetic cause). 

## 3. Discussion

Phosphate entails approximately 0.6% of the total weight of animals and plants [10]. It is considered the second most abundant mineral in the human body, representing about 0.5% of total infant weight [11] and nearly 1% of total weight in adults [10,12,13,14]. It is necessary for many different vital processes, such as energy metabolism, the synthesis of DNA and RNA, and the regulation of proteins by phosphorylation [12,13,15,16,17,18,19]. 

In adults, hypophosphatemia is relatively frequent in daily medical practice [20,21,22,23], but, in contrast, there are few studies assessing the leading causes of pediatric hypophosphatemia [1,13]. In our study, we found that mild hypophosphatemia is not so rare in the pediatric age. It is present in at least 1% of the pediatric patients with any medical condition (268 out of 21,398 studied patients, which corresponds to 1.2%). The reference limits of phosphatemia vary according to the stage of mineralization and growth, and hence with age in children. Therefore, there is no universally accepted definition of chronic hypophosphatemia. It is postulated that the pediatric population needs higher amounts of phosphate due to major requirements for skeletal development and mineralization. However, skeletal development is not linear during infancy and adolescence, and therefore the phosphate demand is not constant either. Thus, several authors have suggested different age-dependent reference ranges of serum phosphate. As an example, Florenzano P. et al. assembled an age-dependent gap, making four different clusters in pediatrics [1]. Koljonen et al. examined serum phosphate concentrations in a large cohort of Finnish children aged 12–24 months and assessed modifying factors that can influence phosphate levels. They concluded that the mean phosphate concentration was 5.9 mg/dL at 12 months in both boys and girls and 4.96 mg/dL when reaching 24 months [13]. 

For the purpose of this study, we used 4 mg/dL as the lower end of phosphate in patients under 12 years of age and 3 mg/dL in those aged 12–16 years as an adequate cut-off point to consider hypophosphatemia, according to what is promulgated. This is in line with the limits proposed in some reference textbooks [24] and is also consistent with the overall distribution of serum phosphate values found in our study population. 

In more than 50% of cases, hypophosphatemia was an intermittent condition; in less than 2% of cases, it was persistent. Unfortunately, in more than 40% of patients with hypophosphatemia, we only have a single analysis of serum phosphate, and therefore the course of the abnormality cannot be determined. In most cases, there was not an underlying condition that would directly explain the hypophosphatemia.

As phosphatemia can be altered by different mechanisms, including inappropriate urinary losses, the diagnostic work-up should include measuring the phosphate urinary excretion. Surprisingly, in our cohort, the renal phosphate losses were assessed in less than 5% of patients with mild but persistent plasma hypophosphatemia. Moreover, in the medical charts, there were no comments about the phosphatemic status. This reflects the fact that in clinical practice, pediatricians often neglect hypophosphatemia and do not adequately study its causes and consequences. This is somewhat disturbing in view of the adverse consequences of hypophosphatemia on the musculoskeletal system and other body organs [1,25,26,27]. However, this observation is similar to others in large cohorts in which vitamin D status was evaluated. Less than 50% of vitamin D studies were accompanied by calcium studies; less than 2% assessed the phosphate status; and less than 1% included PTH [28].

In our cohort, we identified a significant number of patients with different endocrine disorders associated with mild but intermittent hypophosphatemia. We do not know if this metabolic condition alters the endocrine-metabolic equilibrium and has subsequent clinical relevance or if it just reflects a different whole-body phosphate balance. Hypophosphatemia was particularly frequent in patients with diabetes, and serum phosphorus was inversely correlated with serum glucose and glycosylated hemoglobin. Although we did not have data about urine phosphate, hypophosphatemia is likely related to the phosphaturic effect associated with glycosuria more than the potential intracellular shift due to an excess of insulin. Furthermore, reduced tubular reabsorption of phosphate, as a direct effect of acidosis, likely contributes to low phosphate levels in patients with acutely decompensated diabetes [29,30]. In keeping with this, a negative correlation was found between phosphate plasma levels and values of glycosylated hemoglobin, but this correlation disappeared when HbA1c values at the time of the diagnosis of diabetes were excluded from the analysis. This also suggests that phosphatemia correlates more with short-term serum glucose than with the long-term control of diabetes.

Diabetic patients frequently present other electrolyte disorders, mainly hypo- and hypernatremia, hypopotassemia, and hypomagnesemia. These disturbances are more frequent in decompensated patients, especially in the context of diabetic ketoacidosis [31]. Despite an underlying phosphate depletion, phosphate serum levels can be normal or even high in patients at diagnosis [32] because insulin deficiency and metabolic acidosis tend to move phosphate outside the cell. The administration of insulin induces the intracellular shift of glucose and phosphorus into the skeletal muscle and liver cells, leading to hypophosphatemia, especially if there is an underlying situation of phosphate depletion [2,31]. Thus, various studies in adult diabetics with ketoacidosis show an insulin-associated decrease in phosphatemia in up to 74–90% of cases, especially between 8 and 16 h after insulin administration [29,32,33]. In our study, serum phosphate was only measured in two patients during insulin therapy, and therefore the effects of insulin in the pediatric population could not be assessed. 

Apart from phosphate redistribution and urinary losses, low intestinal absorption may also contribute to phosphate depletion in diabetes. In poorly controlled patients, anorexia and malnutrition can be frequent [34], as dietary quality is poor nowadays [35]. Moreover, phosphorus absorption may be impaired by vitamin D deficiency. This is very frequent in patients with diabetes, with a prevalence between 15 and 65% at the end of winter, depending on latitude [36,37,38]. In Spain, there is a lack of specific studies in type 1 pediatric diabetic patients, but in the general pediatric population, deficient vitamin D levels (<20 ng/mL) range from 24 up to 73% in the north of the country [39,40,41]. In our study, only 15% of patients had their vitamin D measured, so we cannot draw any conclusions in this regard. 

Since phosphorus homeostasis in the pediatric diabetic patient is a marker for metabolic control of the disease, we recommend clinicians pay more attention to and study phosphorus metabolism more rigorously in patients with diabetes, with serial analyses of serum and urinary phosphate levels. By knowing the mechanisms underlying phosphorus deficiency, we can take specific measures, in some cases with dietary changes and phosphorus supplements, in others with vitamin D, and in some other cases by optimizing insulin treatment. Moreover, we must not forget the importance of phosphorus in multiple metabolic processes, including glycolysis which uses phosphate to synthesize ATP [42].

Likewise, in children with growth retardation on growth hormone (GH) replacement therapy, plasma and urine phosphate values could also be useful biomarkers. On one hand, chronic hypophosphatemia is associated with growth retardation; on the other hand, exogenous GH has enteral and renal effects that notably influence calcium/phosphate homeostasis [7,9,43,44]. The changes in serum phosphate with GH therapy observed in this study are in line with that concept. 

The major strength of our study is that we reviewed serum phosphate in a large number of pediatric patients with a variety of conditions. However, it also has important limitations related to its largely observational and retrospective design. Consequently, the clinical and biochemical options available were limited in many cases, including repeated phosphate analyses that would help to determine the time course of serum levels and the scarcity of urinary analyses that would help to clarify the causes of hypophosphatemia.

## 4. Patients and Methods

### 4.1. Study Design

The study center was the Hospital Universitario Marqués de Valdecilla. This is a tertiary care center in Cantabria, a region in Northern Spain with a population of 550,000. The hospital serves as the primary hospital for a population of about 320,000 and as the reference center for the rest of the region, which is also served by two community hospitals. The target population, those between 1 and 16 years of age, comprised roughly 82,000 individuals. The study included a retrospective and a prospective arm. In the retrospective arm, we performed a computerized search of the laboratory database between October 2002 and January 2021. We looked for patients with at least one determination of serum phosphorus. 

The prospective arm was carried out between January 2020 and March 2021. Within that time period, a phosphorus test was added to all lab requests from patients who met the above inclusion criteria. At this stage, all lab requests in the region of Cantabria were included. Thus, in addition to the Hospital Marqués de Valdecilla, the community hospitals in Sierrallana and Laredo also participated in the study. This was done by implementing an algorithm within the common system for ordering lab tests.

Patients with hypophosphatemia, defined by serum phosphorus < 4 mg/dL in children aged 1–11 years and <3 mg/dL in those over 11 years (24), were reviewed and identified from both retrospective and prospective arms and their clinical records. Patients with persistent hypophosphatemia were invited to attend our outpatient clinic for clinical evaluation and additional biochemical or genetic tests.

### 4.2. Laboratory Procedures

Serum samples were obtained in the morning after an overnight fast. Serum metabolic variables including creatinine, estimated glomerular filtration rate (2009 Schwartz-IDMS) (eGFR), total calcium, phosphorus, alkaline phosphatase, albumin, and intact parathyroid hormone (PTH) were analyzed using an Atellica CH&IM platform (Siemens Healthcare Diagnostics, Malvern, PA, USA); 25-hydroxycholecalciferol levels were measured using an automated competitive chemiluminescence assay (Liaison XL, DiaSorin Inc., Stillwater, MN, USA).

### 4.3. Data Analysis

All pediatric patients with low serum phosphate were identified at least once. For the purposes of this study, we classified them into the following groups: (a) “persistent hypophosphatemia”, when phosphate levels remained < 4 mg/dL (or <3 mg/dL in subjects over 11 years) in at least two different samples separated by two months, with no subsequent normalization documented; (b) ”isolated hypophosphatemia”, included patients with a single analysis with low phosphate and no subsequent reassessment; and (c) ”intermittent hypophosphatemia”, in the case of patients with both low and normal values of serum phosphate. 

Next, in line with established clinical practice, hypophosphatemia was grouped according to its origin as “expected” (EH) in cases where an underlying disease known to cause hypophosphatemia was present; otherwise, it was considered “unexpected” (UH). Furthermore, the EH group was divided according to the potential pathophysiological mechanisms involved: decreased intestinal absorption, phosphate redistribution, or increased urinary losses.

Data with a normal distribution were expressed as the mean SD; data with a non-normal distribution were expressed as the median IQR. When comparing different groups, the lowest serum phosphate level was used. A correlation and a linear regression model were used for the relationship analysis of the variables. An analysis of variance (ANOVA) test was used for group comparisons, followed by a t-test and Bonferroni adjustment for multiple comparisons. A value of *p* < 0.05 was considered to indicate statistical significance.

### 4.4. Ethical Aspects

The study was approved by the IRB (Comité de Ética de Investigación con Medicamentos de Cantabria, reference 2019.294). Written informed consent was obtained from the parents or legal guardians of patients evaluated at the outpatient clinic.

## 5. Conclusions

In the pediatric age group, malignancies, hypovitaminosis D, and endocrine disorders, mostly diabetes, were the most common causes of hypophosphatemia. This real-world study also shows that hypophosphatemia is frequently neglected and inadequately evaluated by pediatricians, which emphasizes the need for more education and awareness about this condition to prevent its potentially deleterious consequences.

## Figures and Tables

**Figure 1 ijms-24-00687-f001:**
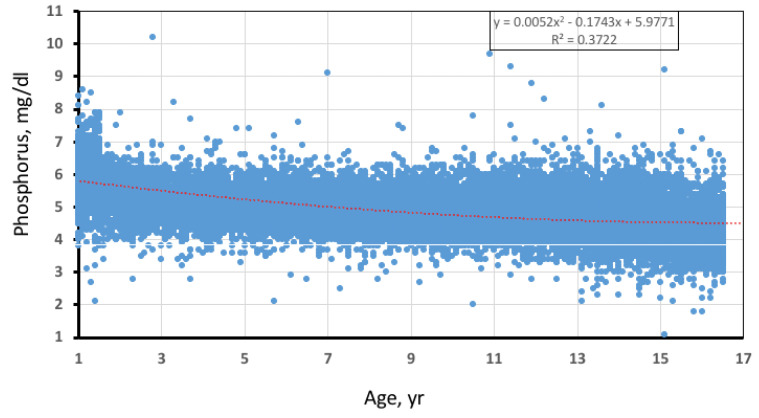
Distribution of serum phosphate results across the pediatric population.

**Figure 2 ijms-24-00687-f002:**
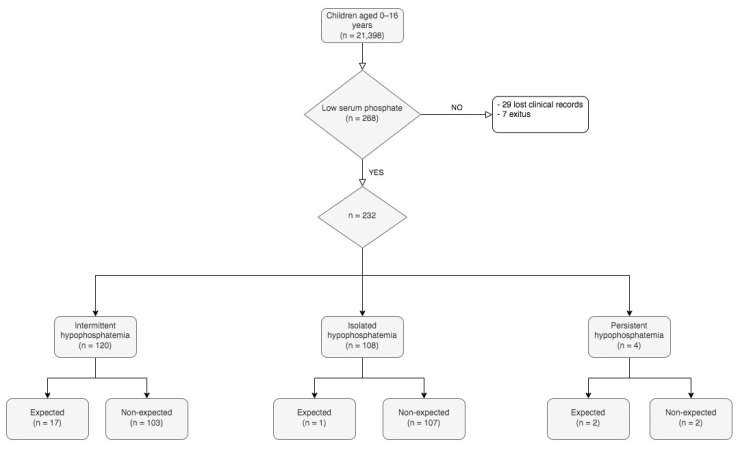
Distribution of cases across subgroups according to the time course and the presence or absence of a hypophosphatemia-causing disorder.

**Figure 3 ijms-24-00687-f003:**
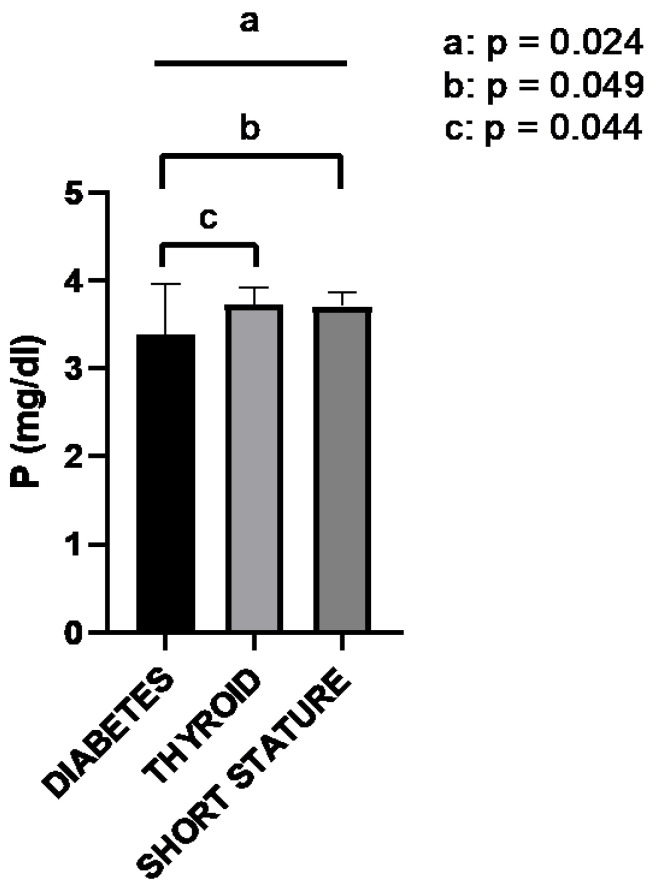
Mean and standard deviation of serum phosphate in children with various endocrine disorders.

**Figure 4 ijms-24-00687-f004:**
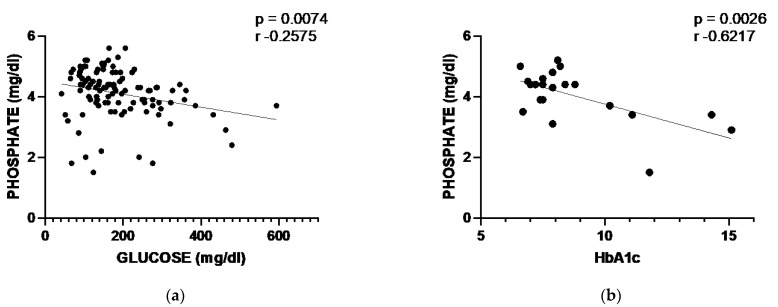
(**a**) Phosphorus-glucose correlation and (**b**) phosphorus-HbA1c correlation.

**Table 1 ijms-24-00687-t001:** Biochemical variables in patients with hypophosphatemia.

Parameter	Reference Interval	Number of Patients	Patients with Abnormal Results (%)	Most Extreme Value ^a^	Mean	Median (IQR)
**Phosphate (mg/dL)**	<12 years: 4–6 ≥12 years: 3–5.4	232	100	1.4	3.6	3.8 (3.5–3.9)
**Albumin (g/dL)**	3.8–5.1	173	17	1.8	4.3	4.4 (4.02–4.7)
**Corrected Ca (mg/dL)**	8.7–10.4	173	12	7.1	9.2	9.2 (8.83–9.4)
**Alkaline phosphatase (U/L)**	1–3 years: 104–345 4–6 years: 93–309 7–9 years: 86–315 10–12 years: 42–362 >12 years: 44–147	197	3 ^a^	11,325 ^a^	328	218 (165–288)
**Estimated Glomerular Filtration Rate (mL/min/1.73 m^2^)**	100–120 ^b^	199	4	15	104	105 (95–120)
**25-hydroxy-vitamin D (ng/mL)**	20–50	36	53	8	20	19 (15–25.5)
**Parathyroid hormone (pg/mL)**	18–88	15	27 ^a^	108 ^a^	49	33 (17–35)

^a^ The lowest value is shown for each variable, except for alkaline phosphatase and PTH, for which the highest value is shown. ^b^ Modified Schwartz formula: height (cm) × 0.413/serum creatinine.

## Data Availability

Raw data are available from authors upon reasonable request.

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
