# Peer review of "Mild Hypophosphatemia-Associated Conditions in Children: The Need for a Comprehensive Approach"

_ijms, 2022, doi:10.3390/ijms24010687_

Round 1

Reviewer 1 Report

This is an interesting study by Pablo Docio, Sandra Llorente-Pelayo and coworkers, and also an important voice for better recognition of phosphate metabolism disorders in children. It indicates that ca. 1% of children requiring a blood test have hypophosphatemia of unclear origin. 

It's a large study, but I would appreciate a bit more information about data analysis and a few improvements to that part of the work. I think it's the weakest point of an otherwise very good study.

1. For instance, it's even unclear which correlation method was used (no mention in Methods, or captions for Fig. 4 and 5).

2. I'm also not sure (it's not explicitly stated) that the paragraph describing data starting at line 181 is referring to the "unexpected phosphatemia" group.

3. The decision to pick the top 3 conditions isn't supported by data about frequencies of other conditions (so we don't know if the number 3 is arbitrary or not).

4. It's unclear what's the point of Fig. 3 - the p-value is barely significant and given low sample sizes I'd consider it unreliable to make any conclusion if P has a different concentration between conditions. 

5. I wonder if the distributions of parameters presented in Table 1. shouldn't be statistically assessed (to improve the role of "Patients with abnormal results" column) - something like a K-S test for each parameter between the hypophosphatemic group and the rest. Do results for Vit. D really stand out from all other conditions? 

(Thinking out loud about points 3 and 4 - I wonder if an enrichment analysis wouldn't fit better the goal that authors want to achieve with respect to describing/understanding the group with unexpected hypophosphatemia. But that's just an idea.)

I don't have any other comments. Study is designed well and results clearly presented.

Author Response

First of all, thank for the comments as a way to improve our manuscript. 

1. For instance, it's even unclear which correlation method was used (no mention in Methods, or captions for Fig. 4 and 5).

Pearson correlation coefficient. Info included in the text, before p          significance.

2. I'm also not sure (it's not explicitly stated) that the paragraph describing data starting at line 181 is referring to the "unexpected phosphatemia" group.

The cluster of endocrinopathies incorporates subjects with clinical conditions that can explain by itself the existence of hypophosphatemia. As an example, 4 patients suffer from adrenal hyperplasia; these candidates are under long-term corticosteroid therapy, which may cause phosphate deprivation, so they are included in “expected hypophosphatemia” group. As a result, endocrinopathies comprise both “expected” and “unexpected” subjects. This is now stated in the text.

3. The decision to pick the top 3 conditions isn't supported by data about frequencies of other conditions (so we don't know if the number 3 is arbitrary or not).

Fraction of other endocrinopathies are now incorporated in the text, to reflect the importance of the “top 3”.

4. It's unclear what's the point of Fig. 3 - the p-value is barely significant and given low sample sizes I'd consider it unreliable to make any conclusion if P has a different concentration between conditions. 

P value of ANOVA test and adjusted and multiple-test adjusted p-values are now given, stating that pair-wise comparisons did not reach multiple test-adjusted threshold of significance.  

5. I wonder if the distributions of parameters presented in Table 1. shouldn't be statistically assessed (to improve the role of "Patients with abnormal results" column) - something like a K-S test for each parameter between the hypophosphatemic group and the rest. Do results for Vit. D really stand out from all other conditions? 

Thank for the comment. Indeed it would be very interesting to compare patients with hypophosphatemia and normalphosphate levels. Unfortunately, we could not perform meaningful comparisons because the incomplete biochemical data of many children with normal serum phosphate.

Reviewer 2 Report

In this manuscript, the authors investigated a large number of phosphate tests to determine the cause of hypophosphatemia in children and explained the significance of more education needed to avoid the disease. This is an interesting study and subsequently deserves a publication after addressing a small issue.

1. Please adjust the caption position from Figure 2 to Figure 5. The captions are usually under the figure. It’s better to combine Figure 4 and Figure 5 and label them as a and b.

Author Response

First of all, thank for the comment as a way to improve our manuscript.

1. Please adjust the caption position from Figure 2 to Figure 5. The captions are usually under the figure. It’s better to combine Figure 4 and Figure 5 and label them as a and b

Caption position adjusted. Figures 4 and 5 combined to be 4a and 4b.